# Screening for Fungicide Efficacy in Controlling Blackleg Disease in Wasabi (*Eutrema japonicum*)

**DOI:** 10.3390/plants12173149

**Published:** 2023-09-01

**Authors:** Yanjun Liu, Changjiang Song, Xin Ren, Guoli Wu, Zihan Ma, Mantong Zhao, Yujia Xie, Yu Li, Yunsong Lai

**Affiliations:** 1College of Horticulture, Sichuan Agricultural University, Chengdu 611130, China; 2021305061@stu.sicau.edu.cn (Y.L.); 2022305089@stu.sicau.edu.cn (C.S.); 2021205031@stu.sicau.edu.cn (Z.M.); 18080011314@163.com (M.Z.); m18875293149@163.com (Y.X.); liyu@sicau.edu.cn (Y.L.); 2Guangyuan Xifu Biotechnology Company, Guangyuan 628000, China; 18054157795@163.com; 3Jiaxing Agricultural and Fishery Technology Promotion Station, Jiaxing 314000, China; glwoo@163.com

**Keywords:** wasabi, mustard sauce, blackleg, fungicide screening, pathogen isolation

## Abstract

Blackleg disease is devastating for wasabi (*Eutrema japonicum*) production, occurring at any time and everywhere within the main production area of the Sichuan Province, China. There have been very few studies on the chemical control of this disease. In this study, we isolated and identified a local popular strain of the pathogen *Plenodomus wasabiae*. The isolated fungus strain caused typical disease spots on the leaves and rhizomes upon inoculation back to wasabi seedlings. The symptoms of blackleg disease developed very quickly, becaming visible on the second day after exposure to *P. wasabiae* and leading to death within one week. We then evaluated the efficacy of ten widely used fungicides to screen out effective fungicides. The efficacy of the tested fungicides was determined through mycelial growth inhibition on medium plates. As a result, tebuconazole and pyraclostrobin were able to inhibit the mycelial growth of *P. wasabiae*, and the most widely used dimethomorph in local production areas produced the lowest inhibition activity (13.8%). Nevertheless, the highest control efficacy of tebuconazole and pyraclostrobin on wasabi seedlings was only 47.48% and 39.03%, respectively. Generally, the control efficacy of spraying the fungicide before inoculation was better than that after inoculation. An increase in the application concentration of the two fungicides did not proportionately result in improved performance. We cloned the full-length sequence of sterol 14-demethylase (*CYP51*) and cytochrome B (*CYTB*) of which the mutations may contribute to the possible antifungalresistance. These two genes of the isolated fungus do not possess any reported mutations that lead to fungicide resistance. Previous studies indicate that there is a significant difference between fungicides in terms of the effectiveness of controlling blackleg disease; however, the control efficacy of fungicides is limited in blackleg control. Therefore, field management to prevent wound infection and unfavorable environmental conditions are more important than pesticide management.

## 1. Introduction

Wasabi (*Eutrema japonicum*), also known as Japanese horseradish, is the plant material of wasabi mustard, which is an essential condiment when eating Japanese sushi. Allyl isothiocyanate, a biologically active hydrolysis product of glucosinolates, contributes to the special pungency of wasabi [1]. Glucosinolates and their derived pungency compounds can be found in many cruciferous vegetables like radish (*Raphanus sativus*) and mustard (*Brassica juncea*) [2]. Many studies have shown that allyl isothiocyanate possesses antibacterial activities and can impact the gut microbiota [3,4]. There are also numerous other health benefits of wasabi, such as improvement of high blood pressure, cholesterol, plasma triacylglycerol [5], and eliminating nitrite salt in food [6]. A wasabi root, which is in fact a swollen stem (rhizome), is used to grate fresh wasabi mustard. Petioles are used to process the mustard paste and make downstream health products. Flower bolts are utilized for making pickled food [7,8].

Although horseradish (*Armoracia rusticana*) and mustard are also used in making mustard sauce, wasabi is considerably more expensive, and is therefore known as “Green Gold”. This is partially due to the special flavor and superlative quality, and partially due to the difficulty of wasabi planting. The seed propagation of wasabi is quite difficult, including seed production, conservation, and germination [9]. Most importantly, wasabi plants require cool and shady conditions and can only be grown in mountainous areas in China with an altitude above 1500 m, mainly in the Sichuan Province and Yunnan Province. The suitable growth temperature for wasabi is 8–20 °C, and the required light intensity is 110 μmol·m^−2^·s^−1^ [10,11]. High temperatures, strong light, and continuous cropping can lead to various diseases, including blackleg disease, phoma leaf spots, phoma petiole spots, root rot, and downy mildew [12]. The pathogen responsible for blackleg disease, phoma leaf spots, and phoma petiole spots is *Plenodomus wasabiae,* which was first isolated and reported by Japanese scientists in 1932.

Blackleg disease is lethal to seedlings, especially when exposed to unfavorable environmental conditions. However, it is not lethal to adult plants. Nonetheless, the disease seriously affects rhizome quality and has a widespread occurrence in the production of wasabi in the Sichuan Province, China. As the pathogen is immersed in the vascular bundles of rhizomes, dark brown whorls appear from the outside to the inside, and finally extended along the vascular bundles, which look like ink infiltration; no decay and softening occurred during the pathogen infection [13]. *P. wasabiae* belongs to *Deuteromycotina*. The latest taxonomic study classified *P. wasabiae* and *P. lingam* as independent subclasses [14]. The morphology of the mycelium of *P. wasabiae* on a potato dextrose agar (PDA) medium was dense, slender, and white [15]. The blackening of rhizomes caused by *P. wasabiae* is related to the activity of polyphenol oxidase and peroxidase [16].

There are no specialized fungicides commercially available for blackleg disease, making the management of this disease difficult in wasabi production. Several control measures are suggested to control the disease: using seedlings instead of tillers, seed disinfection, managing light and temperature, maintaining the field environment, and avoiding continuous cropping [17,18]. Because the pathogen of blackleg disease invades the inner side of rhizomes, the application of chemical fungicides has a less satisfactory effect. In addition, fungi may acquire antifungal resistance from mutations of cytochrome P450 monooxygenase (*CYP51s*) and cytochrome B (*CYTB*), which are the targets of fungicides [19,20]. *CYP51s* encode sterol 14-demethylase, which is the target of sterol demethylation inhibitors (DMIs), and *CYTB* is a mitochondrion gene that is the target of quinone outside inhibitors (DoIs) [21,22].

In this study, we collected and isolated pathogenic fungus from the main areas where wasabi is grown in China, evaluated the inhibition activity and control efficacy of several common fungicides, and screened out fungicides that are effective at controlling blackleg disease.

## 2. Results

### 2.1. Isolation and Identification of the Pathogen of Blackleg Disease

The pathogen of wasabi blackleg was isolated from a diseased rhizome. The blackleg pathogen invades wasabi rhizomes from mechanical wounds of petioles, roots, and tillers (Figure 1A). Vascular bundles play the role of transport so the infection spreads very rapidly. In cases where wounds do not connect with vascular bundles, the infection develops very slowly. After infection, other parts besides vascular bundles, like the pith, also become dark as time progresses. One fungal strain (P1) was isolated and purified from the diseased rhizomes of wasabi. The P1 colony grew rapidly on a PDA plate, covering the entire petri dish after approximately 7 days when placed at 26 °C in darkness (Figure 1B). The aerial mycelium was dense and white at an early stage, but became gray at a later stage.

The isolated fungus was identified through sequence analysis of the internal transcribed spacer (ITS) between 18s rDNA and 28s rDNA. The PCR amplicon had a length of 750 bp, determined by using the universal primer pair ITS1/ITS4 (Figure 1B). The ITS sequences from six colonies were determined and they turned out to be entirely identical, indicating that we identified only one isolate. We then built a phylogenetic tree by using all reported ITS sequences within the genus *Plenodomus* (Figure 1C). The sequences from *Plenodomus wasabiae* and *Phoma wasabiae* (the name of wasabi used before) were used as the reference. The isolated P1 strain shows the closest relationship with isolate *P. wasabiae* CBS120119, with a sequence identity of 100%. The second closest fungus species is *P. biglobosus*, with a sequence identity of 99.40%. We then inoculated the pathogen back into the detached leaves and in vivo rhizomes. The leaves and rhizomes started to develop black spots within 3 days (Figure 1D). After inoculation, the fungus was isolated once again from the infected sites of diseased leaves and rhizomes, and the same ITS sequence was cloned. Finally, the isolated fungus P1 was confirmed to be the pathogenic fungus causing blackleg disease in wasabi.

### 2.2. Symptom Development of Blackleg Disease in Wasabi

After inoculating wasabi seedlings, the incidence on leaves and petioles was described according to the proportion of lesion area and lesion length, respectively. The leaves and petioles developed light-brown lesions at 2 days post-inoculation (dpi), then a faint chlorotic border appeared and the lesions darkened at 4–6 (dpi). Finally, the leaves exhibited chlorosis and wilting (Figure 2A). For petioles, light black spots became visible at 2 dpi at the wounds, and linear lesions appeared at 3–5 dpi. The proportion of lesions increased rapidly between 5 dpi and 6 dpi. Finally, black lesions extended toward both ends of the petioles covering 0.43–4.86% of the petioles, and breaking off at 6 dpi (Figure 2B). Rhizomes were inoculated with *P. wasabiae* and the lesions darkened at 6 dpi (Figure 2C). Compared to leaves and petioles, the development of black spots on rhizomes was relatively slow, at least in terms of appearance (Figure 2D). The expansion speed of lesions on rhizomes accelerated at 5 dpi, and the lesion became completely black at 6 dpi. After inoculation, the pathogen invades the interior of rhizomes and extents along the vascular bundles instead of the epidermis.

### 2.3. Evaluation of Fungicide Efficacy via Fungistasis Testing

The inhibition effects of fungicides were quantified by measuring the diameter of colonies on fungicide-PDA plates. The diameter of colonies on fungicide-free plates could reach more than 7 cm, whereas the colonies grown on the fungicide plates were more or less negatively affected (Table 1). This indicates that all the tested fungicides worked; however, the degree of influence varied greatly, from 13.80% to 100%. Tebuconazole and pyraclostrobin exhibited the highest inhibitory activity, with a 100% inhibition rate on both the 7th and 14th days. After 14 days of cultivation, the antibacterial activities of the other nine fungicides decreased. Dimethomorph had the worst control effect on mycelial growth, and the inhibition index was nearly zero on the 14th day (Figure 3). These results suggest that tebuconazole and pyraclostrobin had the strongest inhibitory effect against *P. wasabiae* and could totally suppress mycelium growth. All the other fungicides had a less satisfactory effect upon fungistasis testing.

### 2.4. Evaluation of Fungicide Efficacy on Live Seedlings

We applied the fungicides 24 h before inoculation (pre-application) or 24 h after inoculation (post-application). Sterile water and dimethomorph were used as the blank control and negative control, respectively. The control efficacy of each fungicide was investigated within 7 days after inoculation.

Following leaf spraying, the control efficacy of dimethomorph (negative control) was only about 12%, which was much lower than that of the tested fungicides (Table 2). The control efficacy of tebuconazole and pyraclostrobin was about 47% and 38%, respectively. Furthermore, there was no difference between the pre-application and post-application of these two fungicides. Fungicides valomyl and pyraclostrobin had a comparable control efficacy of 37.70% and 39.03%, respectively.

In root irrigation, the control efficacy of dimethomorph (negative control) was about 10% (Table 3). Pre-application of tebuconazole exhibited the highest control efficacy at 40.72%. In contrast, post-application was more effective than pre-application for difenoconazole, which achieved a control efficacy of 31.98% following post-application. The fungicides valomyl and pyraclostrobin exhibited comparable control efficacies at 30.02% and 36.67%, respectively. Pre-application appears to be more effective than post-application for pyraclostrobin. Taken together with the results of the fungistasis experiment, tebuconazole exhibits the highest control efficacy, followed by pyraclostrobin.

The optimal application concentration of tebuconazole and pyraclostrobin was further determined to improve fungicide management (Table 4). Both tebuconazole and pyraclostrobin showed a slight increase in control efficacy as the concentration of the fungicides increased. The highest observed control efficacy of tebuconazole was 64.19%, and for pyraclostrobin, it was 59.12%. However, the increase in fungicide concentration did not result in a proportionally better disease control performance. For tebuconazole, there was a noticeable increase in control efficacy when the concentration was increased from 600 to 1200 μg·mL^−1^. Similarly, for pyraclostrobin, the concentration increase from 3000 to 6000 μg·mL^−1^ had a relatively significent impact. The proposed concentrations of tebuconazole and pyraclostrobin are 1200 μg·mL^−1^ and 9000 μg·mL^−1^, respectively, when pesticide residue is not taken into consideration.

### 2.5. Sequence Analysis of the CYP51 and CYTB Genes

The less satisfactory control efficacy of the tested fungicide implies the possibility of antifungal resistance of the isolated *P. wasabiae*. We then cloned the full-length sequences of *CYP51* and *CYTB* from the fungus to examine the possible antifungal resistance mutations. The genes *PwCYP51* and *PwCYTB* were cloned from cDNA by conserved primers which were designed according to the sequences from other fungi within the genus *Plenodomus.* As a result, *PwCYP51* and *PwCYTB* were obtained, which had the lengths of 1782 bp and 1176 bp, respectively. It is reported that the mutations at D134G, Y137F, and K143R of *CYP51* result in antifungal (DMI) resistance, while mutations at G143A and F129L of *CYTB* can cause antifungal (QoI) resistance. However, neither *PwCYP51* nor *PwCYTB* possess these mutations (Figure 4). The low efficacy of the tested fungicides in blackleg disease control is probably due to reasons other than antifungal resistance mutations.

## 3. Discussion

Blackleg disease is highly prevalent in wasabi production; however, the literature relating this disease is very limited. In this study, we isolated the pathogen *P. wasabiae* from the primary production regions in the Sichuan Province, China. The isolation of local *P. wasabiae* may help in disease control in wasabi production. The isolated strain had strong infection ability and caused classical symptoms of the blackleg disease. The disease developed very rapidly and the lesions became visible within 2–3 days after artificial inoculation and the seedling leaves developed chlorosis, wilting, and darkened within one week. The development of black spots on the rhizome was comparatively slower in appearance than on the leaves and petioles. However, this does not imply less severity, as the pathogen infiltrated the rhizome through wounds and extended along the vascular bundle of the rhizome interior, rather than on the epidermis (Figure 1A).

Specific fungicides for blackleg are not yet commercially available. Abuse of fungicides not only pollutes the environment and causes fungicide residues, but also leads to antifungal resistance. In addition, misuse of fungicides does not work in disease management. The current study proposed tebuconazole and pyraclostrobin as the most effective fungicides for the control of blackleg disease. Tebuconazole is a broad-spectrum triazole fungicide used as a seed dressing and foliar spray to control a wide range of diseases such as rusts, smuts, bunt, powdery mildew, leaf spots, and blights, and it can inhibit the synthesis of ergosterol, one of the main components of the cell membrane [23]. In previous studies, tebuconazole had a good preventive effect on apple ring rot, mulberry sclerotiniosis, and tomato root rot [24,25,26]. Tebuconazole is a DMI (sterol demethylation inhibitor) fungicide. DMIs target the enzyme 14α-demethylase which is encoded by the *CYP51* gene [27]. The fungus can benefit from antifungal resistance from *CYP51* mutations [19]. The mutation site Y137F has been reported in resistant strains of many pathogens [21,28]. The mutation D134G in *C. albicans* contributes to resistance against DMIs [29]. Pyraclostrobin belongs to QoIs, of which the fungicidal activity relies on the binding at the so-called Qo site (the outer quinol oxidation site) of the cytochrome *bc_1_* enzyme complex [22,30]. Mutations at G143A, F129L, and G137R produce antifungal resistance [20,31]. However, none of these mutations were observed in *P. wasabiae*.

According to the toxicity test, the inhibitory activity of tebuconazole and pyraclostrobin reached 100%; however, the highest control efficacy was only about 64% with the same concentration in the field environment. In field experiments, factors such as soil composition, temperature, humidity, light, plant growth conditions, seedlings age, and microbial community in the soil can affect the fungicide toxicity, leading to unexpectedly lower control efficacy. With the increase in concentration, the control efficacy of tebuconazole and pyraclostrobin improved, but not significently. Higher concentrations did not help improve the control efficacy. From the perspective of food safety, it is necessary to further explore the pesticide residues of these two pesticides in the prevention and treatment of wasabi blackleg disease. According to the “National Food Safety Standard Maximum Residue Limits (2021)” of China, the residues of tebuconazole and pyraclostrobin in vegetables should not exceed 0.05 mg·kg^−1^ and 2 mg·kg^−1^ for food safety, respectively.

The control efficacy of tebuconazole on *P. wasabiae* was relatively high. Pesticide management before and after disease occurrence in field production was simulated by applying fungicides before and after inoculation. There was no significant difference between the pre- and post-application of fungicides in leaf spraying. Pre-application is significantly more effective than post-application in root irrigation, as roots absorb fungicides in advance which plays a better protective role. These results provide useful information to guide disease management in wasabi production.

## 4. Materials and Methods

### 4.1. Pathogen Isolation of Blackleg Disease

Diseased plants were collected in Guangyuan (E105°89′, N32°60′), at an altitude of 1600 m. The margin between symptomatic and healthy tissues was cut into small pieces (5 by 5 mm). These pieces were surface sterilized in 70% (*vol*/*vol*) ethanol for approximately 20 s and then rinsed in 1% (*wt*/*vol*) sodium hypochlorite for 40 s, followed by rinsing three times with sterilized distilled water. The blackened rhizomes were shredded and then used to draw the line on a potato dextrose agar (PDA) medium. The plates were incubated in the dark at 26 °C and a monoclonal culture was used for pure culture.

### 4.2. Pathogen Identification

The pathogenic fungus of blackleg disease was inoculated onto a PDA medium and incubated at 26 °C for 7 days. The colony morphology, color, and growth rate were observed and recorded [32]. Slides were prepared by picking mycelium with an inoculating needle, and morphological characteristics such as mycelium, color, and pycnidium were observed under optical microscopes.

Genomic DNA was extracted from the lyophilized mycelium of re-isolated strains using the CTAB method, and PCR amplification was performed using universal fungal primers, ITS1/ITS4. The primers were ITS1 (5′-TCCGTAGGTGAACCTGCGG-3′) and ITS4 (5′-TCCTCCGCTTATTGATATGG-3′). The PCR conditions were 94 °C for 5 min; followed by 30 cycles of 94 °C for 40 s, 57 °C for 45 s, 72 °C for 50 s; and a final extension at 72 °C for 10 min. The amplified PCR products were detected via electrophoresis on a 2% agarose gel, and were then sent for sequencing. The homology of sequences was checked with corresponding strains in the GenBank database and used for the identification of fungi. The sequences were analyzed using the nucleotide Basic Local Alignment Search Tool (BLAST). Then, the maximum likelihood trees were constructed using MEGA 7.0 software [33].

### 4.3. Pathogenicity Determination

For ex vivo leaf inoculation, healthy leaves of wasabi were collected in the laboratory. The healthy wasabi leaves were rinsed with sterile water, then disinfected with 75% alcohol for 30 s, rinsed with sterile water three times, and placed on sterile filter paper to dry. Five leaves were placed in each Petri dish. The fungal blocks were punched at the edge of the colony using a sterilized puncher with a diameter of 0.9 cm. Minimally invasive wounds were created on the leaves with a sterilized scalpel, and then the fungal blocks with a diameter of 0.9 cm were punched at the wounds.

Inoculation in live wasabi seedlings: The healthy wasabis were selected and the rhizomes to be inoculated were pricked with sterile inoculation needles. Then, a 0.9 cm mycelial block was inoculated on the wound and the inoculation site was covered with sterile medical gauze. Three replicates were treated, with daily observations and recordings. After seven days, the symptoms were recorded and photographed.

According to Koch’s rule, tissues from the the junction of diseased and healthy leaves and rhizomes were collected for re-isolation and purification. The re-isolated strains were then compared with the original strains, and only those that were consistent with the original pathogenic fungus were used.

### 4.4. The Incidence of Blackleg Disease in Wasabi

The isolated and identified strain was inoculated in live leaves, petioles, and rhizomes of wasabi (70 days old) to investigate the incidence of blackleg disease. The parts to be inoculated were pricked with sterile inoculation needles. Subsequently, a 0.9 cm mycelial block was inoculated on the wound and the inoculation site was covered with sterile medical gauze. Three replicates were treated for each organ.

### 4.5. Evaluation of Fungicide Efficacy via Fungistasis Testing

The toxicity of 10 fungicides was quantified by mycelium growth inhibition. The amount of each fungicide in the PDA medium followed the package instructions (Table 5). The fungicide solution was added to the medium (9.50 mL PDA medium + 0.50 mL fungicide) to assess their inhibitory effects on mycelium growth, while sterile water was added as a control. The medium plates were placed at 26 °C in the dark. The diameter of the fungal plaques for each treatment was measured on the 7th and the 14thdays.

The growth inhibition rate calculation formula is as follows:$$\mathrm{Growth}\;\mathrm{inhibition}\left(\%\right)=\frac{\left(\mathrm{dc}\;-\;1\right)-\left(\mathrm{dt}\;-\;1\right)}{\mathrm{dc}\;-\;1}\times 100$$
dc: colony diameter of the control group; dt: colony diameter of the experimental group.

### 4.6. Evaluation of Fungicide Efficacy on Live Seedlings

To evaluate the control efficacy, live wasabi seedlings (70 days old) were selected, and the leaves were pricked with sterile inoculation needles. Mycelial blocks were then inoculated into the wound. In order to establish an effective method of fungicide application for controlling blackleg disease, we assessed application time (fungicides application before or after the inoculation) and application method (leaf spraying or root irrigation). The concentration of fungicides followed the package instructions. Water and fungicide dimethomorphs were used as the blank control and negative control, respectively. After fungicide application, each treatment involved inoculating 10 wasabi seedlings at three locations for each seedling. The disease classification was measured based on the severity levels (level 1–5). Level 0, no disease; Level 1, disease spots covering less than 5% of the leaf area; Level 2, disease spots covering 6–10% of the leaf area; Level 3, disease spots covering 11–30% of the leaf area; Level 4, disease spots covering 31–50% of the leaf area; Level 5, disease spots covering more than 50% of the leaf area.
$$\mathrm{Disease}\;\mathrm{index}\;\left(\%\right)=\frac{\sum \left(i\times \mathrm{Li}\right)}{n\times L}\times 100$$
i: the severity level of disease; Li: the number of units at severity level grade i; n: the highest-level representative value; L: the total number of units.
$$\mathrm{Relative}\;\mathrm{control}\;\mathrm{efficacy}\;\left(\%\right)=\left(\frac{\mathrm{CK}-\mathrm{PT}}{\mathrm{CK}}\right)\times 100$$

CK: the disease index of the control treatment area; PT: the disease index of the treatment area.

To assay the effect of the fungicide concentration on the disease control, the concentrations of fungicides used were tebuconazole 600 μg·mL^−1^, 1200 μg·mL^−1^, 1800 μg·mL^−1^, 2400 μg·mL^−1^; pyraclostrobin 3000 μg·mL^−1^, 6000 μg·mL^−1^, 9000 μg·mL^−1^, and 12,000 μg·mL^−1^. The fungicides were sprayed 24 h after inoculating *P. wasabiae*.

### 4.7. Sequence Amplification of CYP51 and CYTB from P. wasabiae

The total RNA of *P. wasabiae* was extracted using the EASY spin Plus Plant RNA Kit from Aidlab, and cDNA were obtained by using reverse-transcription Hifair^®^II 1st Strand cDNA Synthesis Super Mix for qPCR from Yesen. To clone the full-length sequence of *CYP51*, the primers positioned at the 5′UTR and the last CDS were designed in the conserved regions of *Plenodomus tracheiphilus* (Genbank: MU006292) and *Leptosphaeria maculans* (Genbank: AY142146). The primer CYP51-P-F was 5′-GGCTTCGCCTCCTTTATCCT-3′ and the primer CYP51-P-R was 5′- CTACTCGACCTTCTCCCTCC-3′. The PCR program was as follows: 35 cycles of 98 °C for 10 s, 57 °C for 30 s, 72 °C for 10 s, and 12 °C for 5 min. To clone the full-length sequence of *CYTB*, primers CYTB-P-F (5′- TGCACTATAACCCTAGTGTAGCAG-3′) and CYTB-P-R (5′- CCATTTATTGAATTTGGTCAAA-3′) were designed according to the conserved regions of *Phoma* sp. (Genbank: OM236666) and *Didymella pinodes* (Genbank: NC_029396). Primers CYTB-P-F and primers CYTB-P-R were positioned at the 5′UTR and 3′UTR, respectively. *Phoma* sp. and *Didymella pinodes* belong to the same family, i.e., *Didymellaceae*. The PCR program was set as follows: 42 cycles of 98 °C for 10 s, 55 °C for 30 s, 72 °C for 5 s, and 12 °C for 5 min. The PCR products were sequenced by Sangong BioInc., Shanghai, China. The sequences were aligned using the software ClustalW 2.0.

## Figures and Tables

**Figure 1 plants-12-03149-f001:**
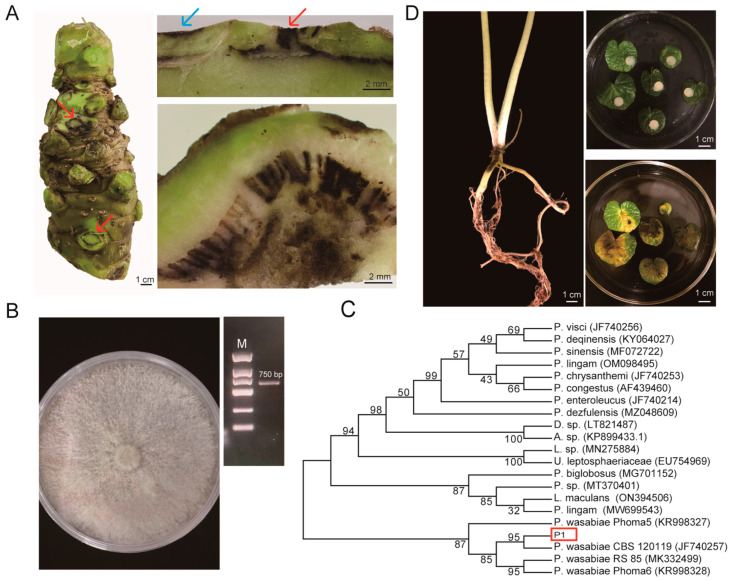
Identification of *P. wasabiae*. (**A**) The disease developed fast along vascular bundles. The red arrows and blue arrows indicate the infected vascular and stem epidermis. (**B**) Morphology of the fungus plaque on a PDA medium and the electrophoresis of ITS sequence. (**C**) Phylogenetic analysis of internal transcribed spacer (ITS) sequences between 18S rRNA and 25S rRNA. P1 in the red box indicates the isolated pathogen in this study. (**D**) The isolated fungus caused blackleg disease when inoculated back to leaves and rhizomes.

**Figure 2 plants-12-03149-f002:**
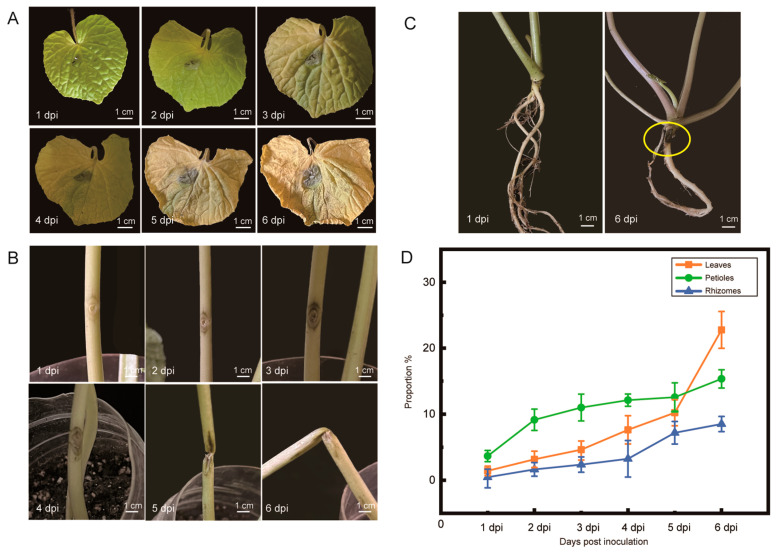
Symptom development of blackleg disease after artificial inoculation. (**A**) Symptoms on the leaves. (**B**) Symptoms on the petioles. (**C**) Symptoms on the rhizomes. The black disease spot was noted by the yellow circle. (**D**) The severity of blackleg disease was quantified according to the proportion of disease spots. The proportion of the area covered by disease spots on the leaves. The proportion of the length covered by disease spots on the petioles and rhizomes. The symptoms were investigated every day from 1 day post-inoculation (dpi) to 6 dpi.

**Figure 3 plants-12-03149-f003:**
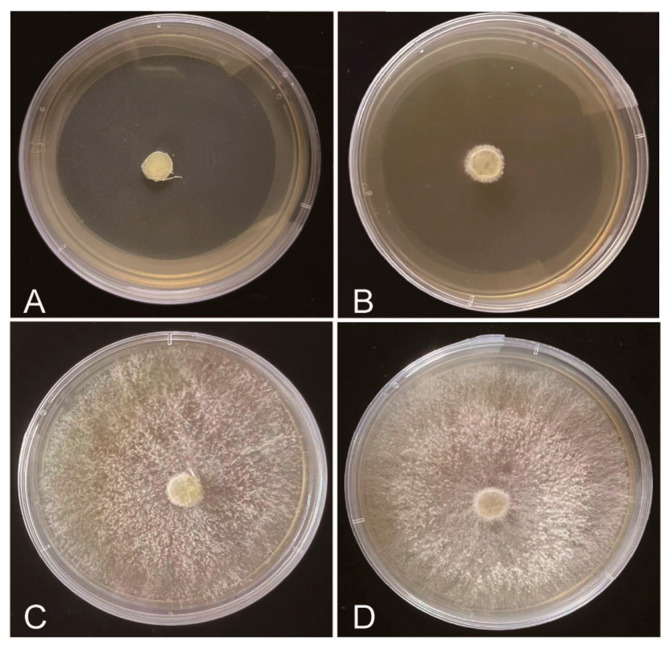
Inhibition activity of the tested fungicides on *P*. wasabiae growth. (**A**) Tebuconazole. (**B**) Pyraclostrobin. (**C**) Dimethomorph. (**D**) CK: 9.50 mL PDA medium + 0.50 mL sterile water.

**Figure 4 plants-12-03149-f004:**
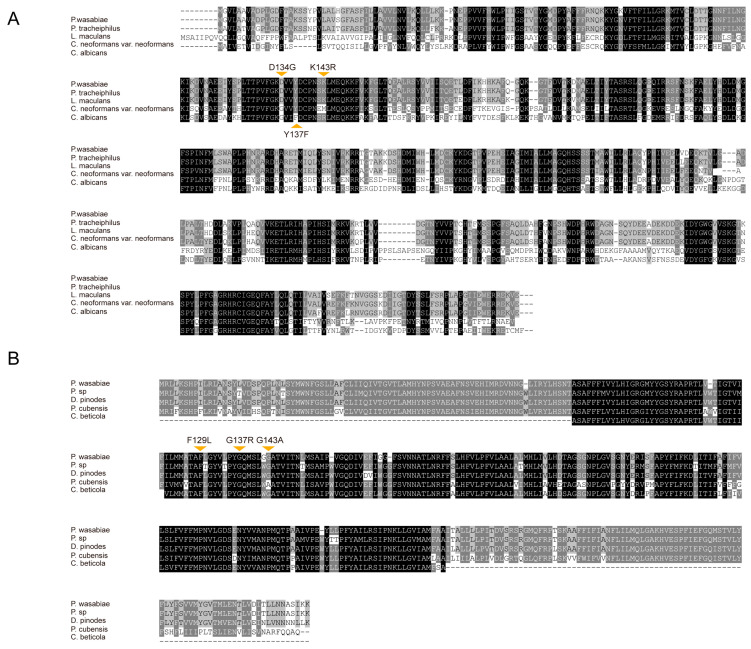
The comparison of *PwCYP51* and *PwCYTB* with homologous genes that possess antifungal resistance mutations. (**A**) *CYP51*. *Leptosphaeria maculans* (Genbank: CBX97082), *Plenodomus tracheiphilus* (Genbank: KAF2854801), *Cryptococcus neofomans var. beoformans* (Genbank: AF225914) and *Candidaalbicans* (Genbank: AF153850). (**B**) *CYTB*. *Didymella pinodes* (Genbank: YP009233081), *Phoma sp.* (Genbank: UXC95304), *Pseudoperonospora cubensis* (Genbank: AHC28174) and *Cercospora beticola* (Genbank: AFN43048).

**Table 1 plants-12-03149-t001:** Inhibition activity of different fungicides against the growth of *P. wasabiae*.

Fungicides	Diameter of Colony (cm) on the 7th Day			Inhibition Index	Diameter of Colony (cm) on the 14th Day			Inhibition Index
	R 1	R 2	R 3		R 1	R 2	R3	
CK	7.60	7.55	7.58	--	8.10	8.20	8.10	--
Dimethomorph	7.30	5.20	7.10	13.80 ± 8.69 e	8.10	8.15	8.10	0.20 ± 0.20 g
Azoxystrobin	5.50	5.60	4.50	31.36 ± 4.66 d	8.00	7.90	8.05	1.84 ± 0.93 g
Valomyl	2.55	2.90	2.50	65.02 ± 1.72 c	3.70	3.75	3.60	54.71 ± 0.42 b
Chlorothalonil	3.60	3.90	3.40	52.04 ± 1.99 c	6.20	6.35	6.05	23.78 ± 0.81 d
Famoxadone	3.80	3.10	3.40	54.70 ± 2.59 c	6.50	6.60	6.50	19.67 ± 0.08 e
Prochloraz	5.50	5.10	5.20	30.49 ± 1.46 d	8.10	8.10	8.10	0.41 ± 0.41 g
Topsin-m	4.20	3.90	4.20	38.86 ± 7.68 d	7.00	6.85	7.10	14.13 ± 1.22 f
Pyraclostrobin	0.00	0.00	0.00	100.00 ± 0.00 a	0.00	0.00	0.00	100.00 ± 0.00 a
Tebuconazole	0.00	0.00	0.00	100.00 ± 0.00 a	0.00	0.00	0.00	100.00 ± 0.00 a
Difenoconazole	1.50	1.50	2.00	78.00 ± 2.20 b	4.80	4.75	4.90	40.77 ± 0.74 c

CK: 9.50 mL PDA medium + 0.50 mL sterile water.Note: Different lower-case letters in the same column represent a significant difference (*p* < 0.05). R1–3, repeats 1–3.

**Table 2 plants-12-03149-t002:** Control efficacy by leaf-spraying of the fungicides.

Fungicide	Spraying before Inoculation		Spraying after Inoculation	
	Disease Index	Control Efficacy	Disease Index	Control Efficacy
CK	65.29 ± 0.42 a	——	65.29 ± 0.42 a	——
Dimethomorph	57.33 ± 1.33 a	12.15 ± 2.63 b	57.57 ± 1.30 ab	11.83 ± 1.97 c
Difenoconazole	58.84 ± 2.36 a	9.85 ± 4.07 b	34.89 ± 0.89 d	46.57 ± 1.22 a
Tebuconazole	34.29 ± 3.30 b	47.48 ± 5.03 a	35.24 ± 0.95 d	46.03 ± 1.31 a
Valomyl	40.67 ± 2.40 b	37.70 ± 3.80 a	47.56 ± 5.78 bc	27.27 ± 8.43 bc
Pyraclostrobin	39.78 ± 3.89 b	39.03 ± 6.17 a	40.81 ± 5.50 cd	37.45 ± 8.53 ab

CK: Spraying with sterile water. Note: Different lower-case letters in the same column represent a significant difference (*p* < 0.05).

**Table 3 plants-12-03149-t003:** Control efficacy following root irrigation of the fungicides.

Fungicide	Root Irrigation before Inoculation		Root Irrigation after Inoculation	
	Disease Index	Control Efficacy	Disease Index	Control Efficacy
CK	66.67 ± 1.28 a	——	66.67 ± 1.28 a	——
Dimethomorph	60.00 ± 2.31 b	10.00 ± 3.00 d	59.52 ± 2.08 a	10.72 ± 2.51 b
Difenoconazole	53.81 ± 1.72 b	19.25 ± 2.64 c	45.43 ± 3.31 b	31.98 ± 3.73 a
Tebuconazole	39.52 ± 2.90 d	40.74 ± 3.97 a	46.67 ± 0.95 b	29.95 ± 1.93 a
Valomyl	46.67 ± 1.33 c	30.02 ± 1.09 b	48.00 ± 2.31 b	27.81 ± 4.86 a
Pyraclostrobin	42.29 ± 2.62 cd	36.67 ± 2.85 ab	48.57 ± 3.30 b	27.19 ± 4.46 a

CK: Root irrigation with sterile water. Note: Different lower-case letters in the same column represent a significant difference (*p* < 0.05).

**Table 4 plants-12-03149-t004:** Control efficacy under different fungicide application concentrations.

Fungicide Name	Concentration (μg·mL^−1^)	Spraying after Inoculation	
		Disease Index	Control Efficacy
CK	——	65.29 ± 0.42 a	——
Tebuconazole	600	35.24 ± 0.95 b	46.03 ± 1.31 b
	1200	26.00 ± 3.06 c	60.15 ± 4.75 a
	1800	30.00 ± 0.00 bc	54.05 ± 0.30 ab
	2400	23.33 ± 3.33 c	64.19 ± 5.37 a
Pyraclostrobin	3000	34.89 ± 0.89 b	46.57 ± 1.22 ab
	6000	27.78 ± 2.22 bc	57.43 ± 3.54 ab
	9000	26.67 ± 3.85 c	59.15 ± 5.86 ab
	12,000	26.67 ± 3.33 c	59.12 ± 5.23 ab

CK: Spraying with sterile water. Note: Different lower-case letters in the same column represent a significant difference (*p* < 0.05).

**Table 5 plants-12-03149-t005:** Information on the ten fungicides used in this study.

Fungicide	Effective Content	Recommended Concentration (g·mL^−1^)	Fungicide Type	Protectant/Systemic
Dimethomorph	10.00%	5000	Morpholines	Systemic
Azoxystrobin	45.00%	2000	Methoxy acrylate	Systemic
Valomyl	66.80%	3000	Amino acid derivative	Systemic
Famoxadone	25.00%	3000	Oxazolone	Protectant
Hymexazol	45.00%	1000	Imidazole	Systemic
Topsin-methyl	50.00%	2000	Benzimidazole	Systemic
Difenoconazole	10.00%	3000	Triazoles	Systemic
Chlorothalonil	38.00%	4000	Aromatics	Protectant
Tebuconazole	30.00%	600	Triazoles	Systemic
Pyraclostrobin	25.00%	600	Methoxy acrylic acid	Systemic

## Data Availability

All the data can be found in the main context.

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
