# Peer review of "Screening for Fungicide Efficacy in Controlling Blackleg Disease in Wasabi (Eutrema japonicum)"

_plants, 2023, doi:10.3390/plants12173149_

Round 1
Reviewer 1 Report
In general, the information provided by authors is new and valuable. In this work, they only identify one isolate. Since they have more isolates, they should analyze more than one isolate, at least three. The molecular identification ITS1 and ITS4 for fungal molecular identification needs to be improved at least with one more gene sequence.
In the fungicide efficacy on plant tissue, why you did not use picnidiospores. This inoculum will cause disease under naturak conditions and could change fungicides´ ability to control this disease.

Author Response
(1) In general, the information provided by authors is new and valuable. In this work, they only identify one isolate. Since they have more isolates, they should analyze more than one isolate, at least three. The molecular identification ITS1 and ITS4 for fungal molecular identification needs to be improved at least with one more gene sequence.
>author response
The reviewer intended to say “isolate/strain” or “colony sequence”? We in fact identified ITS sequence form more than one colony and they were totally the same. There is only one “isolate/strain” identified in this study but we don’t think it is necessary to use many “isolate/ strain” here. Moreover, little is known about property of different “isolate/strain” in P. wasabiae.
We add one sentence:
“The ITS sequences from six colonies were determined which were totally the same indicating the same isolate.” (P3, L97-98)
(2) In the fungicide efficacy on plant tissue, why you did not use picnidiospores. This inoculum will cause disease under natural conditions and could change fungicides´ ability to control this disease.
>author response
There are two reasons that we did not use pycnidiospore. Firstly, we tried many times but failed to observe pycnidiospore in medium culture and we did not find a literature showing the method to produce pycnidiospore in medium culture. In the original version, there was a sentence “Pycnidia were produced on PDA after approximately 12 days”. This is a mistake and we have deleted this sentence. Secondly, we think pycnidiospore is good in root-inoculation but mycelium is better in inoculation on leaves and petioles.
FORMAL ASPECTS
(3) Line 52 and 53. "An unfavorable condition and continuous cropping cause many diseases including Blackleg disease, phoma leaf spot, phoma etiole spot, root rot, downy mildew and so on”. Revise the term "cause", "unfavorable condition and continuous cropping" are not the cause of these diseases. And the term "so on" is vague; please replace it with another word.
>author response
As the reviewer suggested, we replace “cause” by “result in” and we delete “and so on”.
The new sentence is as following:
“High temperature, strong light and continuous cropping result in many diseases including Blackleg disease, phoma leaf spot, phoma petiole spot, root rot, and downy mildew [12].” (P2, L55-57)
(4) Line 62. P. wasabiae is a kind of Deuteromycotina fungus. Revise the term ¡is a kind!
>author response
We changed “P. wasabiae is a kind of Deuteromycotina fungus.” to “P. wasabiae belongs to Deuteromycotina.” (P2, L65)
(5) Line 280. To test the fungal fungicide effect on plates, why do you use "plates containing tap water were used as control"? The control should be only PDA medium plus tap water. Can you explain?
>author response
We agree with the reviewer that it was not our intended meaning. We intended to say “water” instead of “fungicide solution” was added to the medium. Moreover, there is a mistake. We intended to say sterile water.
The new sentence is fowllowing:
“The fungicide solution was added to the medium (9.50 mL PDA medium + 0.50 mL fungicide) to assay their inhibition on mycelium growth, while sterile water was added be control.” (P9, L306-308)
(6) Line 291. Why do you use "blocks of mycelia" instead of conidia? Under natural conditions, this is the way that naturally occurs.
>author response
It is very hard to see conidia in medium culture, although we have tried many times to observe conidia. We quantify the inhibition of fungicides by measuring mycelia growth.
(7) Line 294-295 The concentration of fungicides was recommended concentration. Who recommended?
>author response
The fungicide package suggested the suitable concentration.
We add a new sentence:
“The amount of each fungicide in the PDA medium followed the package instruction (Table 5).” (P9, L304-305)
(8) Line 296-297. After application, Each treatment was inoculated with 10 seedlings and repeated 3 times. Each, why in capital the first letter? What was repeated three times the application or the inoculation process?
>author response
We have changed this error and the new sentence is as following:
“After fungicide application, each treatment was done by inoculating 10 wasabi seedlings at three places for each seedling.” (P10, L321-323)
(9) Line 145. In table 1, CK should be described the meaning at the end of the Table. And in all tables where CK appears.
>author response
We have added the description of CK in the footnote of Table 1.
“CK: 9.50 mL PDA medium + 0.50 mL sterile water.”(P5, L149)
(10) Line 153. Tip water? or tap water was used?
>author response
We have changed “tip water” and “tap water” in the manuscript to “sterile water”.
(11) Line 164. Correct "applicaiton"
>author response
We have corrected the spelling error. (P5, L168)
(12) Line 164-167. "In contrast, post-application is better than pre-application for difenoconazole, which had a control efficacy of 31.98%. Fungicide valomyl and pyraclostrobin had a comparable control efficacy, about 30.02% and 36.67%. Pre-application seems much better than post-application for pyraclostrobin. In summary, tebuconazole has the best control efficacy, followed by pyraclostrobin". According to this, in the end, you should include valomyl.
>author response
We made a conclusion about the fungicide effective according to both mycelium inhibition experiment and inoculation experiment. Tebuconazole and pyraclostrobin could totally inhibit the fungus growth but valomyl could not.

Reviewer 2 Report
Dear authors
I have now read your manuscript and find it quite interesting. I found only few studies on the chemical control Phoma wasabiae in wasabi (Eutrema japonicum). But I must say that the manuscript has a weak point. You have made your conclusion about the sensitivity of the pathogen to tested fungicides testing only one single isolate. I am afraid that the obtained data does not give information about the fungicide resistance in the population of Phoma wasabiae in the region. The azoles and strobilurin have a single target protein. The mutations in these proteins reduce the sensitivity to these fungicides. Based on the data in the manuscript you do not know either you have isolated the wt or mutant. So, as I mentioned already you have no idea what the situation in the population is (in the field or region where you isolated the pure culture).
Because I cannot consider the publication of the manuscript in the present form, but I can only recommend major revision before the manuscript can be considered suitable for publication.
Minor comments/recommendations
1. If possible, include more isolates into your experiments (isolates may originate from other labs, cooperation)
2. If point 1, is not possible, please analyse the target proteins CYP51 and Cyt B for mutations involved in fungicide resistance.
3. Lane 224 -225 – you mention here that root irrigation is one possible way to suppress pathogen. How this is possible in practice in the fields? How do you identify that the plats are infected?
Author Response
(1) I have now read your manuscript and find it quite interesting. I found only few studies on the chemical control Phoma wasabiae in wasabi (Eutrema japonicum). But I must say that the manuscript has a weak point. You have made your conclusion about the sensitivity of the pathogen to tested fungicides testing only one single isolate. I am afraid that the obtained data does not give information about the fungicide resistance in the population of Phoma wasabiae in the region.
>author response
We determined the fungi “isolate/strain” by sequencing multiple colonies (ITS region). Only one “isolate/strain” was identified. We have to point that little is known about property of different “isolate/strain” in P. wasabiae. Our purpose is to test fungicides not to characterize the population genetics of P. wasabiae.
(2) The azoles and strobilurin have a single target protein. The mutations in these proteins reduce the sensitivity to these fungicides. Based on the data in the manuscript you do not know either you have isolated the wt or mutant. So, as I mentioned already you have no idea what the situation in the population is (in the field or region where you isolated the pure culture). Because I cannot consider the publication of the manuscript in the present form, but I can only recommend major revision before the manuscript can be considered suitable for publication.
>author response
The reviewer gave a very professional comment and we appreciate that. The reviewer must be professional in plant pathology. This manuscript focuses on field management and pesticide management in a view of horticulture science rather than plant pathology. We isolated the fungus form major production area of wasabi and then evaluate the fungicide efficacy. So, it does not matter that the fungus is wt or mutant. And we did not pay attention to the genetic structure of the pathogen. That is the object of plant pathologist not a horticulturist. We want to say the literature about P. wasabiae is rare, and there should be other studies to profile the pathogen population structure.
Minor Comments
(3) If possible, include more isolates into your experiments (isolates may originate from other labs, cooperation)
>author response
The literature about P. wasabiae is rare and people don’t know how many races there are for this pathogen. We did not find other race/strain from the same region, although we got the ITS sequences from multiple colonies. The plant wasabi is not a widely-grown vegetable.
(4) If point 1, is not possible, please analyse the target proteins CYP51 and Cyt B for mutations involved in fungicide resistance.
>author response
This is a very good suggestion. We cloned the full-length sequence for CYP51 and CYTB from P. wasabiae. Sequence analysis indicates that the isolated fungus does not possess any antifungals-resistance related mutations.
(5) Lane 224 -225 – you mention here that root irrigation is one possible way to suppress pathogen. How this is possible in practice in the fields? How do you identify that the plants are infected?
>author response
As horticulturists, we have to say root-irrigation is not a hard task. There are many ways like agricultural sprayer (spray around root), and drip irrigation. An infected plant is not a prerequisite for pesticide application. It is unnecessary to confirm every infected plant before fungicide application.

Reviewer 3 Report
Experiment planning and preparation are thorough, even professional (Use of reisolated and sequenced strains, etc.). Texting: characterized by a nice and concise style. The results confirm the set goals, and moreover, the results open a window for new research (207-208).
Errors recommended for correction.
Latin names were not highlighted: 98, 101, 149, 312, 324, 325, 326, 343, 366,
Highlighting foreign language parts: 102, 254
ad 103-104: …The pathogenic parts…. (?)
ad 294-295. (proposed:...was around the recommended concentration..)
ad 153 Correct: ..tap water…
ad 164. Correct: ..application ….
ad table 2-3-4 These were left without serious explanations.
ad fig 2/D Statistics and partially the comment left unexplained.
Only some sentences are difficult to understand.
See my comments in the "Comments and Suggestions for Authors
".
Author Response
(1) Latin names were not highlighted: 98, 101, 149, 312, 324, 325, 326, 343, 366,
>author response
We have made all these Latin names italic.
(2) Highlighting foreign language parts: 102, 254
>author response
We have made all these Latin names italic.
(3) ad 103-104: …The pathogenic parts…. (?)
>author response
The meaning of “The pathogenic parts” is the place where inoculated with the pathogene and appeared symptom.
We changed this sentence:
“The fungus was then isolated again from the infection site of the diseased leaves and rhizomes after inoculation and the same ITS sequence was cloned.” (P3, L105-107)
(4) ad 294-295. (proposed:...was around the recommended concentration..)
>author response
We have changed this sentence, and the new one is “The concentration of fungicides followed the package instruction.” (P10, L321-322)
(5) ad 153 Correct: ..tap water…
>author response
We have changed “tap water” to “sterile water”. (P5, L157)
(6) ad 164. Correct: ..application ….
>author response
We have changed this spelling error, “application”(P10, L321-322 ).
(7) ad table 2-3-4 These were left without serious explanations.
>author response
We have added or detailed the foot note of table2/3/4.
(8) ad fig 2/D Statistics and partially the comment left unexplained.
>author response
We have added the error bar and significant differnce in Fig2D.

Round 2
Reviewer 2 Report
Dear authors
I am satisfied with the changes you have made to the manuscript. I hope that you will agree with me that the present version of the manuscript is much better than the first one.
Good luck with your further research!